# Extracorporeal Magnetotransduction Therapy as a New Form of Electromagnetic Wave Therapy: From Gene Upregulation to Accelerated Matrix Mineralization in Bone Healing

**DOI:** 10.3390/biomedicines12102269

**Published:** 2024-10-07

**Authors:** Lennart Gerdesmeyer, Jutta Tübel, Andreas Obermeier, Norbert Harrasser, Claudio Glowalla, Rüdiger von Eisenhart-Rothe, Rainer Burgkart

**Affiliations:** 1Department of Orthopaedics and Sports Orthopaedics, Klinikum rechts der Isar, Technical University of Munich, Ismaninger Str. 22, 81675 Munich, Germany; 2ECOM Excellent Center of Medicine, Arabellastraße 17, 81925 Munich, Germany; 3BG Unfallklinik Murnau, Professor-Küntscher-Straße 8, 82418 Murnau am Staffelsee, Germany

**Keywords:** electromagnetic wave therapy, bone regeneration, matrix mineralization, osteoblastogenesis, fracture healing, non-invasive therapy

## Abstract

Background: Electromagnetic field therapy is gaining attention for its potential in treating bone disorders, with Extracorporeal Magnetotransduction Therapy (EMTT) emerging as an innovative approach. EMTT offers a higher oscillation frequency and magnetic field strength compared to traditional Pulsed Electromagnetic Field (PEMF) therapy, showing promise in enhancing fracture healing and non-union recovery. However, the mechanisms underlying these effects remain unclear. Results: This study demonstrates that EMTT significantly enhances osteoblast bone formation at multiple levels, from gene expression to extracellular matrix mineralization. Key osteoblastogenesis regulators, including SP7 and RUNX2, and bone-related genes such as COL1A1, ALPL, and BGLAP, were upregulated, with expression levels surpassing those of the control group by over sevenfold (*p* < 0.001). Enhanced collagen synthesis and mineralization were confirmed by von Kossa and Alizarin Red staining, indicating increased calcium and phosphate deposition. Additionally, calcium imaging revealed heightened calcium influx, suggesting a cellular mechanism for EMTT’s osteogenic effects. Importantly, EMTT did not compromise cell viability, as confirmed by live/dead staining and WST-1 assays. Conclusion: This study is the first to show that EMTT can enhance all phases of osteoblastogenesis and improve the production of critical mineralization components, offering potential clinical applications in accelerating fracture healing, treating osteonecrosis, and enhancing implant osseointegration.

## 1. Introduction

Bone consists of approximately 60% mineral (inorganic), 30% organic material, and 10% water, with the organic component predominantly comprised of type I collagen (COL1A1) [1]. Research has consistently demonstrated that the collagen matrix acts as a scaffold for bone mineral deposition [2]. The collagen-rich extracellular matrix (ECM) undergoes mineralization through the deposition of hydroxyapatite [3]. Chemically, the inorganic material comprises ionic calcium (Ca^2+^) and inorganic phosphate (Pi). Pyrophosphate (PPi), consisting of two inorganic Pi groups, is one of the most potent mineralization inhibitors [4]. Therefore, the hydrolysis of PPi is critical for the mineralization process. Alkaline phosphatase (ALP), often regarded as a marker for bone metabolism, plays a pivotal role in regulating ECM mineralization [5]. Moreover, phosphatase orphan 1 (PHOSPHO1) and nucleoside pyrophosphohydrolase-1 (NPP1), encoded by the *ENPP1* gene, are additional vital PPi/Pi ratio regulators, modulating the mineralization process [6].

Osteoblasts are the primary bone-forming cells. Besides ALP and a significant amount of type I collagen, osteoblasts produce osteocalcin (OCN). OCN and its gene (*BGLAP*) are widely trusted indicators in scientific research for measuring osteoblastic activity [7].

Osteoblast differentiation, termed osteoblastogenesis, progresses through three phases: proliferation, matrix maturation, and mineralization [8]. The transition of osteoprogenitors from proliferation to ECM maturation is orchestrated by RUNX2, leading to the expression of bone-related genes. Simultaneously, Osterix (*SP7*) plays a guiding role in osteoblast differentiation [9].

Constituting over 95% of all bone cells, osteocytes play a central role in regulating bone formation and mineralization [10]. Specific proteins, uniquely expressed in osteocytes, serve essential functions in maintaining phosphate homeostasis. These include phosphate-regulating genes with *PHEX* (phosphate-regulating neutral endopeptidase on chromosome x) and *MEPE* (matrix extracellular phosphoglycoprotein) [10].

Since the initial discovery of a potential piezoelectric effect in bones in 1957, extensive research has been devoted to investigating the modulation of bone formation through the application of electric fields [11]. Following approval by the Food and Drug Administration (FDA) in 1979, numerous devices have been developed to produce a distinct form of electromagnetic wave therapy known as pulsed electromagnetic field (PEMF). Demonstrating promising potential in enhancing bone repair, electromagnetic wave therapy has been effectively utilized to address various osteogenic disorders. This includes the successful treatment of fractures, particularly non-union fractures, as well as mitigating bone loss linked to osteoporosis or radiation exposure [12,13]. Although some studies propose that PEMF influences osteoblastogenesis and cell proliferation, the effectiveness of PEMF therapy in influencing bone formation remains unclear and inconclusive [14]. Intracellular calcium transients are widely recognized as crucial in regulating osteoblast proliferation and differentiation, thereby playing a fundamental role in bone formation. Numerous studies suggest that calcium is pivotal in enhancing osteoblastogenesis triggered by magnetic field exposure [13,15].

Most PEMF devices can generate magnetic field strengths ranging from 1 mT to 10 mT [16]. A recent advancement in electromagnetic field therapy is Extracorporeal Magnetotransduction Therapy (EMTT), which sets itself apart from PEMF by using high-intensity electromagnetic fields with a strength reaching up to 150 mT. These physical parameters facilitate a more substantial biological impact, promising a new therapeutic approach to managing bone disorders [17]. Since EMTT is a relatively novel form of electromagnetic wave therapy, there is limited available evidence regarding its effectiveness. Yet, a few case reports have demonstrated enhanced bone healing [18,19]. EMTT has been successfully used to treat non-unions and lower back pain [18,20]. Clinical practitioners have characterized it as a significant advancement in treating complex bone disorders and chronic inflammatory conditions [17].

To date, only a single cellular study has been conducted, offering potential insights into the osteogenic effects of EMTT. The in vitro analysis was performed on human bone marrow mesenchymal stem cells (MSCs) and revealed increased vascular endothelial growth factor (VEGF) levels and upregulation of bone formation-specific genes such as *COL1A1* and *ALP* [21]. These findings imply the potential benefits of EMTT in bone fracture healing through the modulation of bone metabolism and angiogenesis mechanisms.

For the first time, this study represents a cellular investigation, providing essential biological insights into the observed acceleration of bone formation through EMTT stimulation. We specifically examined the effects of EMTT stimulation on the proliferation, differentiation, and mineralization of primary human osteoblasts (hOBs).

To ensure the clarity of our study’s conclusions, we rigorously maintained identical experimental conditions throughout. This involved employing consistent stimulation and cultivation protocols, including standardized EMTT physical parameters (level 8, 8 Hz, 30 min).

## 2. Materials and Methods

### 2.1. Cell Culture

The isolation of human osteoblasts (hOBs) from human femoral heads followed standard protocols [22]. Culturing was performed in Dulbecco’s Modified Eagle Medium (DMEM, PAN-Biotech, Aidenbach, Germany), supplemented with 0.25% Primocin (InvivoGen, San Diego, USA), 15% fetal calf serum (FCS, Sigma-Aldrich^®^, St. Louis, MO, USA) 1% L-Glutamine (Merck, Darmstadt, Germany) and 20 mM hydroxyethyl-1-piperazinethanesulfonic acid (Hepes, Thermo, Waltham, MA, USA). Additionally, 50 µg/mL ascorbic acid (Merck, Darmstadt, Germany) and 10 nM dexamethasone (Merck, Darmstadt, Germany) were introduced with every medium change. Cells from passages 3 to 5 were selected for experimentation. To induce differentiation and mineralization, the DMEM medium was replaced with Minimum Essential Medium Eagle (Alpha modification, Sigma-Aldrich^®^, St. Louis, MO, USA), supplemented with 10 mM ß-glycerophosphate (Merck^®^, Darmstadt, Germany). The cells were seeded at the specified concentrations. A 6-well plate was used for PCR experiments, with each well containing 100,000 cells (10,416 cells/cm^2^). For the WST-1 assay, a 96-well plate was used with 5000 cells/well (15,625 cells/cm^2^). In the other experiments, a 24-well plate was used, with each well seeded with 35,000 cells (18,421 cells/cm^2^). Half of the conditioned medium was replaced according to our stimulation protocol (Figure 1b). To mitigate any potential variations, only osteoblasts from the third to sixth passage were utilized. Additionally, consistent quantities of fetal calf serum (FCS) and other supplements were employed in the medium throughout all experiments conducted.

### 2.2. Stimulation Protocol

The stimulation protocol remained consistent throughout all experiments and was selected for its clinical applicability. Stimulation sessions were conducted 2–3 times per week, adhering to the schedule outlined in Figure 1b. Parameters were chosen to mirror clinical settings: a duration of 30 min, an intensity level of 8, and a frequency of 8 Hz. An exception was made to capture early gene expression responses to EMTT stimulation. For the PCR analysis on day 3, cells were stimulated on days 1 and 3, allowing just one day between stimulations. Subsequent PCR analyses at days 7 and 14 followed the standard stimulation procedure 2–3 times per week.

**Figure 1 biomedicines-12-02269-f001:**
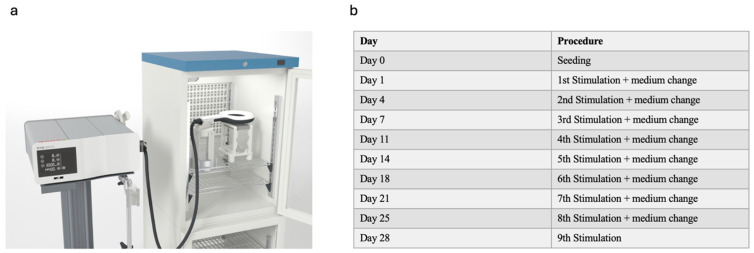
Experimental setup and protocol. (**a**) Experimental setup used in our study. The control and stimulated groups were placed identically in the incubator with the EMTT device turned on for the stimulated group and kept off for the control group. (**b**) Stimulation protocol used in our study. The day of stimulation and medium change are illustrated.

The well plate was placed in a 3D-printed construction with the EMTT applicator positioned above it (Figure 1a). To maintain a stable environment, a 3D-printed door was used to seal the incubator, and the cable was routed through a designated channel to ensure a consistent CO_2_ concentration of 5% and a temperature of 37 °C. Both the experimental and control groups were exposed to these identical conditions. The only difference was that the EMTT applicator was activated for the experimental group, while it remained deactivated for the control group for the same duration. Before each medium change and experimental procedure, microscopic examination was conducted to verify cell viability, ensuring the reliability of subsequent analyses.

### 2.3. Cell Proliferation and Viability

The proliferation rate of hOBs was evaluated using tetrazolium salt WST-1 (Roche, Basel, Switzerland). The WST-1 assay was used to measure the metabolic activity of the cells on days 0 (pre-first EMTT stimulation), 7, and 14 of the culture. Then, 10 μL of WST-1 reagent was added to the wells. After a 3 h incubation at 37 °C, the absorbance was measured using a microplate reader (Multiskan Ascent, Thermo Fisher Scientific, Waltham, MA, USA) at 450 nm, with correction at 620 nm. The hOBs were cultured for 7 and 14 days, and their viability and cytotoxicity were assessed using the LIVE/DEAD^®^ Viability/Cytotoxicity Kit (Invitrogen, Thermo Fisher Scientific, Waltham, MA, USA). The assay is based on the enzymatic conversion of calcein AM to intensely fluorescent calcein within live cells, resulting in a green fluorescence signal (excitation/emission ~495 nm/~515 nm). EthD-1 demonstrates a 40-fold fluorescence enhancement upon binding to nucleic acids within deceased cells, resulting in a red fluorescence signal (excitation/emission ~495 nm/~635 nm). This method relies on the identification of live cells through the presence of intracellular esterase activity, while damaged membranes distinguish dead cells. GFP 488 was employed to visualize Calcein, and TR 594 was used for EthD-1 labeling. Imaging was performed using a Zeiss Observer Z1 microscope (Carl Zeiss AG, Oberkochen, Germany).

### 2.4. Calbryte

Osteoblasts were seeded at a density of 5000 cells per well in a 96-well plate overnight. Optical/black-bottom plates were utilized to minimize light scattering and reduce background fluorescence. To measure calcium levels, we employed Calbryte™ 520 AM (AAT Bioquest, Sunnyvale, CA, USA) at a concentration of 4 µM along with Pluronic F127 (10%) and Probenecid (25 mM). Before dye-loading, the growth medium was replaced with fresh HHBS buffer to prevent interference with the working solution. The working solution was then incubated with the cells for 45 min at 37 °C. Following incubation, the dye-working solution was replaced with HHBS to remove excess probes. Subsequently, the well plates were returned to the incubator. Cells were either stimulated with EMTT stimulation for 10 min or left unstimulated (control) for 10 min. Fluorescence was measured using a fluorescence microscope (Zeiss Observer Z1 microscope) with a FITC filter and a plate reader (Ex: 485 Em: 538). Images captured by the microscope were analyzed and quantified using Image J software (Image J 1.5.4.).

### 2.5. RNA Extraction and qPCR Analysis

Total RNA was extracted from hOBs according to the manufacturer’s protocol (RNeasy^®^ Mini Kit, QIAGEN, Hilden, Germany). The A260/A280 ratio evaluated RNA quality, and only RNA samples with an A260/A280 ratio between 1.8 and 2.1 were selected. The cDNA was synthesized according to the manufacturer’s protocol (QuantiTect Reverse Transcription Kit, QIAGEN). Real-time quantitative PCR was performed on a StepOne™ Real-time PCR system (Applied Biosystems, Waltham, MA, USA) using TaqManTM Assays (ThermoFisher Scientific, Waltham, MA, USA) according to the manufacturer’s protocol. The level of endogenous mRNA was normalized to the level of *GAPDH* mRNA using the 2^−ΔΔCT^ method. The TaqManTM assays used in this study are listed in the Appendix A.

### 2.6. ELISA Protein Detection

The Human Pro-Collagen-1-α1 ELISA Kit (R&D Systems, Minneapolis, MN, USA) was utilized to quantify Pro-Collagen-1-α1, indicating collagen 1 synthesis in hOBs. In a 96-well plate coated with a monoclonal antibody specific for Human Pro-Collagen-1-α1, 50 µL of diluted osteoblast medium underwent a 2 h incubation. After a thorough wash to eliminate unbound substances, an enzyme-linked monoclonal antibody specific for Human Pro-Collagen-1-α1 was added to the wells. A substrate solution was introduced after an additional hour of incubation and washing. The color development reaction continued until reaching the desired intensity, and optical density was measured using a microplate reader (Multiskan Ascent, Thermo Fisher Scientific) at 450 nm, with correction at 560 nm. The final protein concentration was determined by employing a constructed 4-PL standard curve.

The Human Osteocalcin Instant ELISA (Thermo Fisher, Waltham, MA, USA) was utilized to quantify human OCN production in hOBs. Supernatants of osteoblast medium were diluted and added to a 96-well plate coated with the anti-human OCN monoclonal antibody immobilized onto microwells, incubating for 2 h. After washing, an HRP-conjugated monoclonal anti-human OCN antibody was introduced, forming a colored product proportional to the soluble OCN present. The color development reaction continued until reaching the desired intensity. The optical density was measured using a microplate reader (Multiskan Ascent, Thermo Fisher Scientific) at 450 nm, with correction at 620 nm. The obtained absorbance values correlated with a standard curve, enabling calculation of the final OCN concentration.

### 2.7. Stainings

#### 2.7.1. Alizarin Red

Following 4, 7, 11, 14, 21, and 28 days of cell culture, hOBs underwent fixation in a 4% paraformaldehyde solution. Subsequently, cells were stained with a 1% Alizarin Red S solution (Sigma-Aldrich^®^, St. Louis, MO, USA) at pH 4.2 for 10 min, followed by a thorough wash with distilled water. Imaging of the representative nodules was carried out using a Zeiss Observer Z1 microscope. To quantitatively assess calcium deposition, stained samples were treated with 10% cetylpyridinium chloride (Sigma-Aldrich^®^, St. Louis, MO, USA) for 10 min, and absorbance was measured at 405 nm using a microplate reader (Multiskan Ascent, Thermo Fisher Scientific).

#### 2.7.2. Von Kossa

After 4, 7, 11, 14, 21, and 28 days of cell culture, hOBs were fixed using a 4% paraformaldehyde solution. Staining was performed using the Von Kossa Method for Calcium Kit (Polysciences, Warrington, PA, USA), wherein 3% silver nitrate was added to the wells and incubated for 20 min. After washing with distilled water, the staining was fixed in a 5% sodium thiosulfate solution for 2 min and rewashed. A Zeiss Observer Z1 microscope (Carl Zeiss AG, Oberkochen, Germany) was used to image representative nodules, and an overview was created.

#### 2.7.3. Alkaline Phosphatase Staining

After 4, 7, 11, 14, 21, and 28 days of cell culture, hOBs were fixed using a Methanol-Acetone solution (1:1). Wells were rehydrated for 10 min in Tris-HCl, 0.1 M, pH 7.0 (Sigma-Aldrich^®^, St. Louis, MO, USA). The excess solution was removed, and the staining solution was prepared by dissolving one tablet of NBT/BCIP (Roche, Basel, Switzerland) in 10 mL distilled water. After an incubation time of 20 min, the staining solution was removed, and the cells were washed with Tris-HCl, 0.1 M, pH 7.0 (Sigma-Aldrich^®^).

#### 2.7.4. Sirius Red Staining

After 4, 7, 11, 14, and 21 days of cell culture, hOBs were fixed using a Bouin’s solution composed of picric acid, acetic acid, and formaldehyde. The cells were rinsed five times with distilled water. Following air-drying, a 0.1% Sirius Red solution was added to the wells, and the cells were incubated for 1.5 h. After this incubation period, the cells were washed with 0.01 N hydrochloric acid (HCl). Representative nodules were imaged using a Zeiss Observer Z1 microscope. To quantify collagen deposits, the stained samples were treated with 0.1 N sodium hydroxide (NaOH) for 10 min, and absorbance was measured at 570 nm using a microplate reader (Multiskan Ascent, Thermo Fisher Scientific).

#### 2.7.5. Image J Quantification

In our study, ImageJ software was employed to quantify the intensity of cell staining. Digital images of the stained cells were first captured using a Zeiss Observer Z1 microscope. These images were then imported into ImageJ, where a standardized thresholding process was applied to distinguish the stained areas from the background, and the intensity was measured [23].

### 2.8. ALP Kinetic

The ALP activity was evaluated through a colorimetric kinetic assay. The cell culture medium was initially removed, and phosphate-buffered saline (PBS, Sigma-Aldrich^®^, St. Louis, MO, USA) was used to wash the cells. Subsequently, Triton (Triton™ X-100, Sigma-Aldrich^®^, St. Louis, MO, USA) was introduced into the wells and diluted with water to a 5% concentration, leading to cell lysis through repeated freeze and thaw cycles. The assay adhered to the manufacturer’s protocol (Alkaline phosphatase FS DGKC, DiaSys, DiaSys, Holzheim, Germany). ALP dephosphorylates p-nitrophenyl phosphate (pNPP) to p-nitrophenol, generating a yellow color with maximal absorbance at 405 nm. The absorbance values are directly proportional to the enzyme activity within the wells. For kinetic activity determination, measurements were captured at 0 min and 15 min at 405 nm using the Multiskan Ascent spectrophotometer (Thermo Fisher Scientific). Concentrations were then calculated through a standard curve established from p-nitrophenol standards. Pipetting was executed in triplicate, and the mean value was derived.

Concurrently, the total protein concentration from the same well was measured using the Pierce™ BCA Protein Assay Kit (Thermo Fisher Scientific, Waltham, MA, USA), aligning with the manufacturer’s protocol. Absorbance readings were taken at 560 nm using the Multiskan Ascent spectrophotometer (Thermo Fisher Scientific), and concentrations were determined by referencing a generated standard curve. Again, pipetting was performed in triplicate, and the mean value was computed. The ultimate measurement of enzyme activity was quantified as micromoles of p-nitrophenol generated per minute per milligram of the protein.

### 2.9. EMTT Physics

Extracorporeal Magnetotransduction Therapy (EMTT) is an innovative approach for treating musculoskeletal disorders by using pulsed electromagnetic fields. EMTT systems typically feature a decaying sinusoidal magnetic field with an oscillation frequency between 100 kHz and 300 kHz and a field strength reaching 80 mT (Figure 2a,b). We measured the electromagnetic wave strength for each well on a standardized well plate. The central wells exhibited a magnetic wave strength of 80 mT, while the marginal wells showed a slightly lower strength of 74 mT (Figure 2a). To ensure consistent treatment conditions and mitigate any potential limitations due to varying magnetic field strengths, we selected the central wells for our experiments. The transmitter coil, characterized by high inductance (L) and low resistance (R), generates the magnetic field, penetrating biological tissue relatively undisturbed. This field’s spatial distribution can be calculated using Biot–Savarts law, determined by the coil geometry and current flow. EMTT’s mode of action may involve the induction of electrical fields, generation of force (e.g., Lorentz force, torque on magnetic structures), excitation or modulation of excited states (e.g., Zeeman effect, radical pair), and potential thermal effects. However, the primary mechanism likely involves the magnetic field’s rapid oscillation and high amplitude, generating a strong electrical field by induction. This transduction power, exceeding 60,000 T/s for EMTT, is thought to drive therapeutic effects.

### 2.10. Statistics

Statistical analysis was conducted using GraphPad Prism v5.0 software (GraphPad Software Inc., La Jolla, CA, USA). Results are presented as mean ± SD to depict both central tendency and variability within the data. To ensure the reliability of the findings, experiments were performed in triplicate at a minimum. Data were subjected to analysis using the Student’s *t*-test or one-way ANOVA, followed by appropriate post hoc testing. Paired comparisons were assessed using a two-tailed *t*-test, while the paired *t*-test was selectively applied to evaluate related samples. The significance threshold was set at a *p*-value below 0.05, providing a rigorous standard for determining statistical significance.

## 3. Results

### 3.1. EMTT Stimulation Has No Impact on Cell Viability and Proliferation of hOBs

We utilized live/dead viability/cytotoxicity staining to assess cell viability (Figure 3a–e) and the WST-1 reagent to measure proliferation rates (Figure 3f). The live/dead staining method allowed us to distinguish between live (green) and dead (red) cells, providing accurate insights into cellular health [24]. Additionally, the WST-1 assay quantifies cellular metabolic activity and proliferation [25]. Our findings revealed no adverse effects of EMTT stimulation on cell viability or proliferation rates. Live/dead staining analysis showed similar proportions of live and dead cells between stimulated and control groups on days 7 and 14 (Figure 3e). Similarly, the WST-1 assay demonstrated consistent metabolic activity and proliferation levels at days 0, 7, and 14 (Figure 3f).

### 3.2. EMTT Triggers Calcium Influx in hOBs

Calcium ions are fundamental as secondary messengers in numerous cellular processes, including cell proliferation, differentiation, and apoptosis [26]. Understanding calcium dynamics is crucial, as alterations in calcium signaling have been implicated in various physiological and pathological conditions. Calcium signaling has emerged as a critical regulator of osteogenic processes, influencing bone formation and remodeling [27]. The elucidation of calcium-mediated signaling pathways offers potential insights into the mechanisms underlying osteogenic effects, such as those induced by electromagnetic transduction therapy (EMTT). Chosen for its improved resistance to external interferences, Calbryte 520 AM emerged as the preferred Ca^2+^ indicator [28]. Our imaging analysis revealed distinct changes in cellular fluorescence before (Figure 4a) and after EMTT stimulation (Figure 4b). Representative images captured before and after a 10 min stimulation with EMTT demonstrated an increase in fluorescence intensity, particularly in selected cells marked with red arrows, indicative of elevated intracellular calcium concentrations post-stimulation. ImageJ software was utilized for the quantitative analysis of fluorescence intensity, confirming increased calcium concentration following stimulation (Figure 4c). Additionally, fluorescence measurements obtained via a fluorescence plate reader demonstrated significantly higher fluorescence levels in stimulated cells compared to control cells (Figure 4d).

### 3.3. EMTT Stimulation Increases the Production of Osteogenesis-Related Proteins and Genes in hOBs

*RUNX2*, *SP7*, and *SPP1* (Osteopontin) are crucial genes involved in osteogenesis, with distinct roles in bone formation [4,10,29]. RUNX2 acts as a master regulator of osteoblast differentiation, while osterix (*SP7*) regulates the final stages of osteoblast differentiation. We assessed their gene expression levels using PCR (Figure 5a–c). The results revealed significant differences in gene expression levels, with *SP7* peaking significantly on day 3 and showing more than seven-times-higher expression than the control group. *RUNX2* peaked on day 3, with a fourfold increase compared to the control. *SPP1* was significantly upregulated at every measured day, also peaking at day 3. At day 14, the differences observed in gene expression were notably reduced, with *SP7* and *RUNX2* showing no significant differences compared to the control group.

Traditionally, OCN has served as a bone formation marker and mineralization regulator. Recent research has unveiled its broader role as a hormone involved in global homeostasis [30]. Nonetheless, given its status as one of the most abundant proteins, exclusively synthesized by fully differentiated osteoblasts, the protein and its gene (*BGLAP*) are reliable indicators of osteoblastic activity [8]. Our study detected OCN protein levels using ELISA, while *BGLAP* gene expression was detected through PCR analysis (Figure 5d,f). ELISA analysis for OCN revealed higher concentrations in the stimulated group every time after stimulation initiation, showing significant differences on days 4, 7, and 14 (Figure 5f). *BGLAP* gene expression peaked significantly on day 3, with no notable differences observed on days 7 and 14 (Figure 5d).

ALP plays a crucial role in bone metabolism by facilitating mineralization, making it a widely utilized marker for osteoblast-driven bone formation [31]. By hydrolyzing PPi, ALP reduces its concentration and increases the amount of Pi, which is essential for hydroxyapatite formation. We investigated the impact of EMTT stimulation on ALP concentration through ALP-specific staining, PCR analysis for gene expression, and kinetic ALP measurements. ALP staining revealed a substantial concentration of ALP, reaching a plateau by day 14 in both control and stimulated cells (Figure 5g), prompting us to limit the quantification of image staining intensity until that time. The greatest difference was observed at day 4, with decreasing statistical significance noted from day 7 onward (Figure 5h). Nonetheless, a discernible discrepancy in staining intensity persisted. Kinetic measurements demonstrated the highest difference in ALP activity on days 2 and 4, gradually diminishing but remaining significant by day 7 (Figure 5i). Consistent with these findings, PCR analysis showed a notable upregulation of *ALPL* gene expression on days 3 and 7, with the highest significance observed on day 7 (Figure 5e). However, no significant difference persisted by day 14.

### 3.4. EMTT Stimulation Enhances the Collagen Synthesis in hOBs

Collagen plays a pivotal role in bone metabolism, serving as the main structural protein in the extracellular matrix of bone tissue [3]. To assess collagen metabolism, we employed multiple techniques. We utilized Sirius Red staining to visualize collagen deposition within the bone matrix, ELISA to quantify protein synthesis, and PCR to analyze RNA expression levels [32]. We observed the most pronounced differences in Sirius Red staining between day 7 and 21, which diminished at day 28 (Figure 6a). The absorbance measurements of the eluted wells and the quantification of staining intensity revealed the most significant differences at day 14 (Figure 6b,c). ELISA protein detection demonstrated peak significance in collagen synthesis between days 4 and 14, becoming insignificant by day 16 (Figure 6e). The PCR analysis revealed a substantial upregulation of the *COL1A1* gene expression at days 3 and 7, whereas this effect diminished by day 14 (Figure 6d).

### 3.5. EMTT Stimulation Enhances Mineralization and Upregulates Mineralization-Related Genes in hOBs

Our study investigated the impact of EMTT stimulation on mineralization processes in hOBs. Mineralization is a fundamental process in bone formation, with calcium and phosphate deposition forming the mineral component of the bone matrix [33]. To assess mineralization, we utilized Alizarin Red S staining, which visualizes calcium ions, and Von Kossa staining, which detects phosphate ions [34]. The results demonstrated enhanced mineralization in EMTT-stimulated hOBs, as evidenced by increased staining intensity in both Alizarin Red S (Figure 7a,c,e) and Von Kossa staining (Figure 7b,d). Additionally, the expression of mineralization-related genes was analyzed. PHOSPHO1 and ENPP1–3 are crucial regulators of the PPi/Pi ratio, modulating the balance between inorganic pyrophosphate (PPi) and phosphate ions (Pi) to influence mineralization [7]. Among the ENPP family, ENPP1 is the most studied member due to its significant involvement in various physiological processes, including bone mineralization and vascular calcification [35]. MEPE plays a critical role in phosphate homeostasis and bone mineralization, while PHEX regulates phosphate levels in the body, impacting mineralization processes [10]. The gene expression analysis revealed significant upregulation of mineralization-related genes, including *MEPE*, *PHEX*, *ENPP1*–*3*, and *PHOSPHO1*, particularly during the initial phase of treatment (days 1 and 3) following EMTT stimulation (Figure 7f,g). By day 14, these gene expression differences diminished as the treatment progressed (Figure 7h).

## 4. Discussion

Electromagnetic field therapy finds extensive application across various medical domains, including oncology, depressive disorder, fibromyalgia, and osteoarthritis [36,37,38]. Its primary application lies in bone healing, leading to substantial research efforts evaluating the effects of magnetic wave therapy on bone metabolism through in vivo and in vitro studies. Given the influence of physical stimuli on bone healing, magnetic wave stimulation has emerged as a popular and effective method for enhancing bone repair [39]. Several studies have been conducted using hOBs, the critical regulators of bone formation, to understand the molecular pathways. Notably, a recent investigation explored the effect of PEMF stimulation on radiation-induced bone loss, demonstrating its efficacy in mitigating bone loss by enhancing osteoblastic differentiation and mineralization [13]. By applying specific waveforms, they observed Ca^2+^ oscillations. Other studies propose that the effect of electromagnetic field therapy is likely mediated by the opening of voltage-gated calcium channels (VGCCs). This calcium influx can stimulate nitric oxide production through calcium/calmodulin-dependent nitric oxide synthases, potentially contributing to therapeutic responses via the cGMP and protein kinase G pathway, ultimately enhancing bone formation [40]. In our study, EMTT stimulation increased calcium levels, as evidenced by calcium imaging. While the specific mechanism leading to this increase requires further exploration, the significant rise in measured calcium levels following EMTT stimulation underscores the crucial involvement of calcium as a central mediator of EMTT effects.

The impact of electromagnetic wave stimulation on proliferation, differentiation, and mineralization is comprehensively discussed [14]. Chang et al. observed an effect of PEMF stimulation on osteoblast proliferation, yet no effect on cellular differentiation and mineralization. Their findings suggest that studies reporting a stimulated increase in bone tissue-like formation are likely associated with an increase in cell numbers rather than enhancing osteoblast differentiation.

Assessed through a WST-1 assay, the EMTT stimulation showed no significant effect on the proliferation rate at days 7 and 14. Consistently, a study investigating the impact of EMTT stimulation on MSCs also reported no effect on proliferation [21]. Studies across diverse cell types consistently reveal an inverse correlation between proliferation and differentiation processes [41]. Given this biological principle, it can be inferred that the clinically observed accelerated bone healing by EMTT stimulation may be attributed to enhanced differentiation and mineralization rather than the proliferation of hOBs.

Our study shows that EMTT enhance bone formation by influencing various phases of osteoblastogenesis. RUNX2, often referred to as the master regulator of early osteoblast differentiation, plays a crucial role in osteoblastogenesis [10]. Osterix (*SP7*) serves as a transcriptional regulator for the final stages of bone tissue formation, activating *BGLAP* and *COL1A1* genes [42]. *RUNX2* and *SP7* genes exhibited significant upregulation at days 3 and 7. By day 14, this difference became nonsignificant compared to the control. These findings suggest that EMTT stimulation may accelerate differentiation, particularly during the initial stages of treatment.

The organic material is 90% mainly built-up of collagen type 1 (*COL1A1*) [3]. In our study, stimulated hOBs demonstrated a notable increase in collagen production, as evidenced by Sirius Red staining and pro-collagen type 1 ELISA, compared to the control group. Given the minor elevation in Osterix gene expression after 14 days of treatment, it is reasonable to observe that the expression of the *COL1A1* gene was no longer significantly elevated compared to the control group by day 14.

The ECM mineralization primarily consists of hydroxyapatite, the structural foundation of bone, comprising calcium and phosphate [43]. Alizarin Red staining visualizes calcium ions, while Von Kossa staining detects phosphate ions. Our mineralization staining results show that EMTT stimulation initiates mineralization processes earlier than the control group. Remarkably, cells subjected to EMTT exhibit significantly higher levels of hydroxyapatite components—calcium and phosphate—particularly during the initial 14 days of treatment. The early positive effect of EMTT stimulation on mineralization is supported by the observed elevated gene expressions of mineralization-related genes (*MEPE, PHEX, ENPP 1-3*, *PHOSPHO1*) at the early phase of treatment (day 3). Like the other results, the gene expression did not exhibit a relevant difference from the control group by the end of the treatment (day 14).

These findings show that EMTT treatment positively influences both the organic and mineral components of bone, emphasizing its potential for enhancing bone quality. Our results indicate that EMTT stimulation likely accelerates the early phases of osteoblastogenesis. This is demonstrated by ALP and OCN production, which are markers of osteoblast differentiation and activity. With osteoblasts’ maturation, ALP and OCN concentration tends to decrease [33]. At the onset of EMTT treatment, *BGLAP* and *ALPL* expression exhibited upregulation. No significant disparity in gene expression levels was observed throughout the treatment duration, suggesting an initial phase of heightened osteoblast activity followed by a transition to a mature state. This underscores that the effect of EMTT stimulation appears to be more pronounced at the start of treatment, with subsequently diminishing effects.

Our study revealed significant upregulation in the expression of crucial bone-related genes, including *COL1A1*, *SPP1*, *BGLAP*, *ALPL*, and *BGLAP*, particularly during the initial stages of cultivation. This upregulation coincided with the increased expression of pivotal regulators like *RUNX2* and *SP7*. These findings highlight the early molecular responses induced by EMTT stimulation, suggesting a potential mechanism underlying the observed clinical effect of enhanced bone healing.

Despite the promising findings that indicate enhanced bone formation, there are persistent criticisms and concerns regarding the effectiveness of PEMF treatment for bone disorders [44]. One notable critique is the inconsistency of results, potentially stemming from physical factors such as the low achievable impulse frequency or magnetic field intensity of most PEMF devices, which may result in a lower biological impact. Additionally, there is no consensus on the optimal duration of stimulation, with in vitro treatments ranging from 8 min to 24 h over 1 to 28 days and in vivo treatments varying from 1 to 8 h over 1 to 12 weeks [45].

In contrast, the EMTT device utilized in our study (STORZ Medical Magnetolith) generates a magnetic field strength of 80 mT. It operates at an effective transduction power exceeding 60 kT/s, with an oscillating frequency of 100–300 kHz. These physical parameters likely contribute to a more substantial biological impact, reducing treatment time. To assess the possibility of any biological harm to the cells, we conducted live-dead staining after 7 and 14 days of treatment. In conjunction with the proliferation assays, the results demonstrate that EMTT stimulation does not adversely affect cell viability.

As a relatively new form of electromagnetic field therapy, the clinical application of EMTT still needs to be fully documented in the existing literature. Concerning bone disorders, studies have reported an improved healing of non-unions and fractures [18,19]. The typical return-to-sport time following a clavicle fracture is approximately 13.7 weeks [46]. However, there was a notable case where a patient with a lateral clavicle fracture showed rapid recovery [19]. Surprisingly, just three weeks after surgery, the athlete participated in a high-level triathlon following treatment with EMTT, highlighting the effectiveness of this intervention. Other documented indications include inflammatory conditions such as rotator cuff tendinopathy and nonspecific low back pain [20,47]. Given that the observed effects of EMTT on osteoblastogenesis are particularly prominent at the onset of treatment, it can be inferred that EMTT stimulation may be effectively employed clinically to initiate and expedite bone formation. Potential applications include the treatment of bone fractures and enhancement of implant osseointegration. Accelerating osseointegration with surrounding bone would enhance the stability of implants/prostheses and improve the clinical outcomes of various regenerative and prosthetic therapies in orthopedics and dentistry. Although several in vitro and in vivo studies have demonstrated enhanced osseointegration with PEMF stimulation, the lengthy treatment sessions have hindered PEMF therapy from becoming a standard adjuvant therapy [48].

Determining the optimal physical parameters of PEMF stimulation for enhancing bone formation remains a topic of discussion. Li et al. (2021) investigated the efficacy of PEMF stimulation in bone consolidation during distraction osteogenesis in rats [49]. They compared the effects of a traditional FDA-approved PEMF signal with a novel high slew rate (HSR) PEMF signal. Surprisingly, their findings revealed that one hour of HSR PEMF treatment daily provided comparable benefits in bone formation to three hours of traditional PEMF treatment. This highlights that an increase in oscillation frequency results in faster and more effective bone formation.

Considering the expressed concerns regarding the prolonged treatment sessions associated with PEMF devices, EMTT presents a promising alternative. Unlike PEMF devices, EMTT devices operate at higher physical parameters, delivering improved bone metabolism effects within a significantly shorter timeframe. This highlights a crucial advantage of EMTT for clinical application, offering expedited therapeutic outcomes.

As osteoblasts represent just one facet of the complex process of bone formation, further research is warranted to explore the involvement of other cell types, such as osteoclasts, endothelial cells, fibroblasts, and mesenchymal stem cells.

Comprehensive investigations into the underlying biological pathways and cascades are necessary to elucidate the fundamental biological effects of EMTT stimulation on different cell types.

In conclusion, this study is the first to clarify the bone healing effects of EMTT observed in clinical settings. For the first time in magnetic wave therapy research, we have identified a specific waveform that has shown a significant positive impact on all stages of osteoblastogenesis and mineralization, especially during the initial stages. These findings highlight the potential of EMTT as a beneficial, safe, and time-saving treatment for various bone-related conditions, such as fracture healing, osteonecrosis, and implant osseointegration.

## Figures and Tables

**Figure 2 biomedicines-12-02269-f002:**
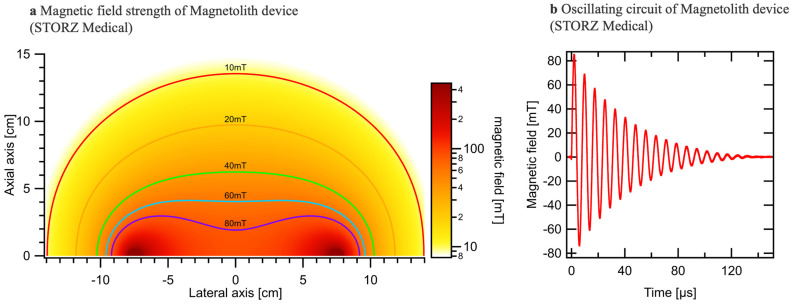
Illustration of the physical parameters of EMTT. (**a**) Variation in the measured magnetic field strength relative to the lateral and axial axis. (**b**) Oscillation frequency reaching up to 300 kHz.

**Figure 3 biomedicines-12-02269-f003:**
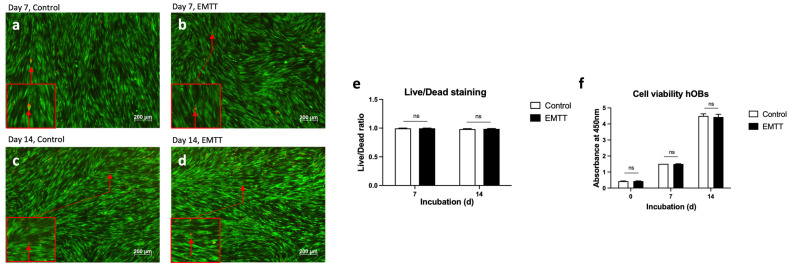
EMTT stimulation has no impact on the cell viability. (**a**–**d**) The hOBs were stained using the live/dead viability/cytotoxicity kit on days 7 (**a**,**b**) and 14 (**c**,**d**). Four representative images are illustrated. Stained with calcein AM (viable cells stain green) and ethidium homodimer 1 (non-viable cells stain red; see red arrow). (**e**) Quantification of living and dead cells was conducted based on counting cells of three wells. Each well is represented by five image excerpts, totaling 15 representative sections (5 per well). (**f**) The measured absorbance at 450 nm (corrected with 620 nm) represents the proliferation rate of hOBs using the WST-1 reagent. Cells were stimulated with EMTT according to our stimulation protocol (Figure 1b). Data are expressed as the average ± SD of three independent experiments (*n* = 3). ns: non-significant. two-tailed Student’s *t*-test.

**Figure 4 biomedicines-12-02269-f004:**
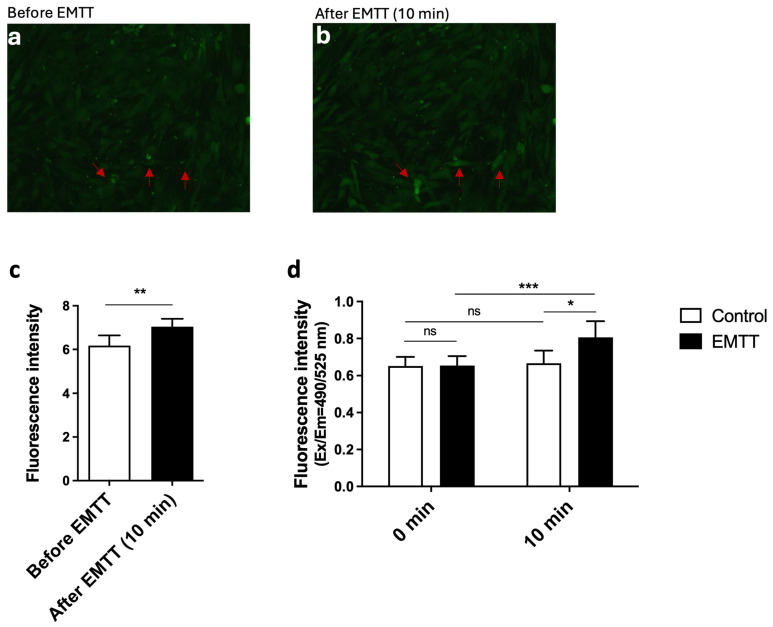
EMTT increases the intracellular calcium concentration in hOBs. The hOBs were incubated at 37 °C for 45 min with Calbryte 520 AM containing buffer in the presence of 2.5 mM probenecid. (**a**,**b**) Cells were stimulated with EMTT for 10 min, and the images were taken before (**a**) and after (**b**) stimulation with a fluorescence microscope using the FITC channel. Two representative images are illustrated. Red arrows show calcium increase in the cells. (**c**) The cellular fluorescence images were quantitated using ImageJ software. (**d**) Fluorescence was measured using a fluorescence plate reader (Ex: 485/Em: 538). Data are expressed as the average ± SD of five independent experiments (*n* = 5). ns: non-significant. * *p* < 0.05, ** *p* < 0.01, and *** *p* < 0.001, paired Student’s *t*-test and two-tailed Student’s *t*-test.

**Figure 5 biomedicines-12-02269-f005:**
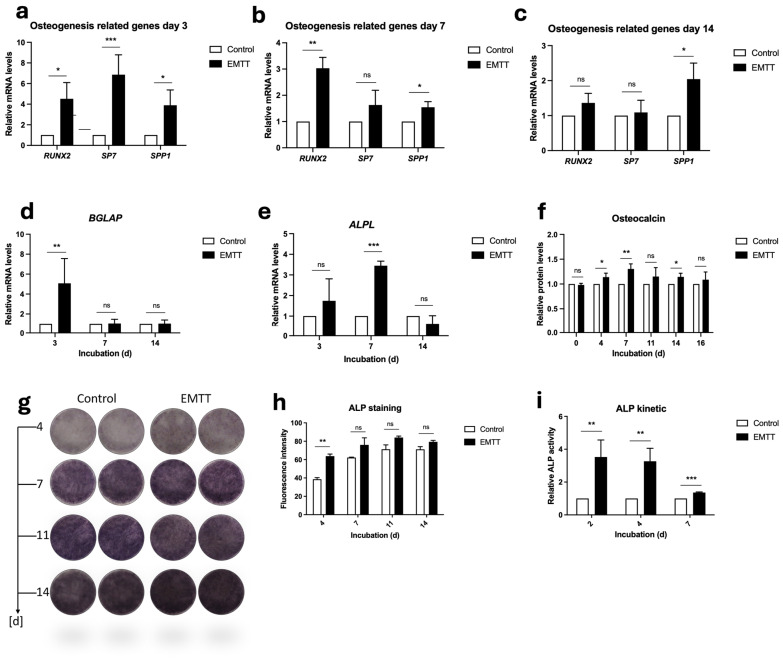
EMTT stimulation enhanced OCN and ALP synthesis and induced an upregulation in osteogenic genes in hOBs. (**a**–**e**) Data analysis was performed by using the 2^−ΔΔCT^ method. Gene expression was normalized to GAPDH and compared by setting control cultures to 1 as a reference value. Cells were stimulated with EMTT according to our stimulation protocol (Figure 1b), followed by RNA extraction and PCR. (**f**) OCN Protein levels were determined using ELISA according to the manufacturer’s instructions. Supernatants were collected at different time points: day 0 (pre-first EMTT stimulation), 4, 7, 11, 14, and 16. (**g**) ALP staining. Two representative wells are illustrated. (**h**) The images were quantitated using ImageJ software. (**i**) ALP activity was measured through a colorimetric kinetic assay and compared by setting control cultures to 1 as a reference value. Cells were stimulated once until day 2, twice until day 4, and thrice until day 7. Data are expressed as the average ± SD of three to six independent experiments (*n* = 3–6). ns: non-significant. * *p* < 0.05, ** *p* < 0.01, and *** *p* < 0.001, two-tailed Student’s *t*-test.

**Figure 6 biomedicines-12-02269-f006:**
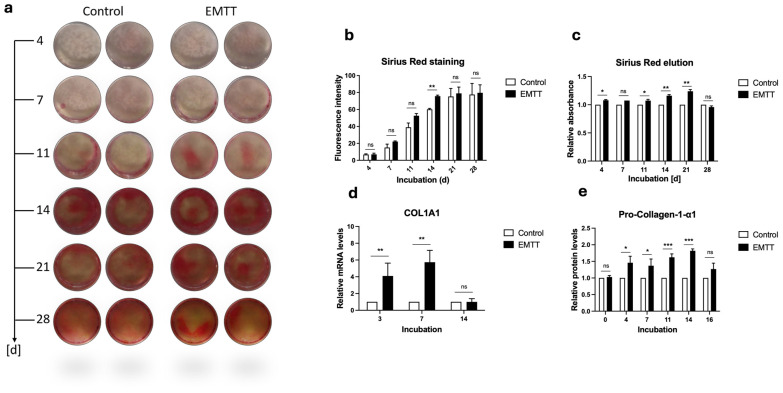
EMTT stimulation enhanced collagen synthesis in human osteoblasts (hOBs). (**a**) Sirius Red staining. Two representative wells are illustrated. (**b**) The images were quantitated using ImageJ software. (**c**) The Sirius Red dye was eluted using 0.1 N Sodium Hydroxide (NaOH), and the absorbance was measured at 570 nm. Results are presented relative to the control (normalized to 1). (**d**) Data analysis was performed by using the 2^−ΔΔCT^ method. Gene expression was normalized to GAPDH and compared by setting control cultures to 1 as a reference value. Cells were stimulated with EMTT according to our stimulation protocol (Figure 1b), followed by RNA extraction and PCR. (**e**) Pro-Collagen-1-α1 protein levels were determined using ELISA according to the manufacturer’s instructions. Supernatants were collected at different time points: day 0 (pre-first EMTT stimulation), 4, 7, 11, 14, and 16. Results are presented relative to the control (normalized to 1). Data are expressed as the average ± SD of three to six independent experiments (*n* = 3–6). ns: non-significant. * *p* < 0.05, ** *p* < 0.01, and *** *p* < 0.001, two-tailed Student’s *t*-test.

**Figure 7 biomedicines-12-02269-f007:**
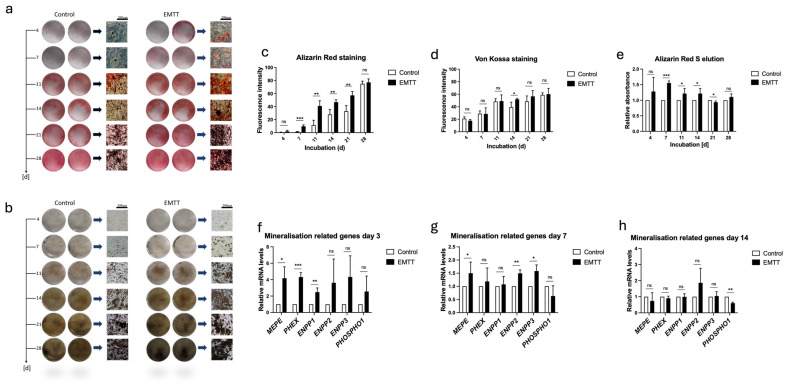
EMTT stimulation enhanced mineralization and upregulated mineralization-related genes in human osteoblasts (hOBs). (**a**) Alizarin Red S staining. (**b**) Von Kossa staining. Two representative wells are illustrated. (**c**,**d**) The images were quantitated using ImageJ software. (**e**) The Alizarin Red dye was eluted using cetylpyridinium chloride, and the absorbance was measured at 405 nm. Results are presented relative to the control (normalized to 1). (**f**–**h**) Data analysis was performed by using the 2^−ΔΔCT^ method. Gene expression was normalized to GAPDH and compared by setting control cultures to 1 as a reference value. (**f**–**h**) Cells were stimulated with EMTT according to our stimulation protocol (Figure 1b), followed by RNA extraction and PCR. Data are expressed as the average ± SD of three to six independent experiments (*n* = 3–6). ns: non-significant. * *p* < 0.05, ** *p* < 0.01, and *** *p* < 0.001, two-tailed Student’s *t*-test.

## Data Availability

All data associated with this study are present in the paper. More data supporting this study’s findings are available from the corresponding author on request.

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
