# Peer review of "Extracorporeal Magnetotransduction Therapy as a New Form of Electromagnetic Wave Therapy: From Gene Upregulation to Accelerated Matrix Mineralization in Bone Healing"

_biomedicines, 2024, doi:10.3390/biomedicines12102269_

Round 1

Reviewer 1 Report

Comments and Suggestions for Authors

The authors describe the effects of cycles of electromagnetic waves in bone healing. In particular, they studied the effects in coltured osteoblast cells detecting an over expression of some genes related to the osteoblastogenesis and bone mineralization. The manuscript appears well written and the results convincing. The introduction and methods are wide and well described. The figures are of good quality and useful. The references updated.

The biological effects detected after electomagnetic wave  can be useful in the bone healing after traumatic events such as fractures. This study has been conducted in coltured cells and in a future work, the authors can demonstrate the same effects in fractured bones of living animals and humans. 

I think that this manuscript deserves to be published in the present form. 

The authors  

Author Response

Dear Reviewer,

Thank you for your positive feedback and supportive comments. We appreciate your suggestion for future research in living models and are glad you found the study convincing and well-presented. We look forward to building on this work in future studies.

Sincerely,

L. Gerdesmeyer

Reviewer 2 Report

Comments and Suggestions for Authors

Comments:

Gerdesmeyer et al. have studied EMTT as a new form of electromagnetic wave that up-regulates the bone-specific gene expression for accelerated bone regeneration. The manuscript is well-organized and well-written. Before possible publication please address the minor comments:

1.     Fig. 3(e and f), increase the font size for better visibility.

2.     Fig. (a and b), provide magnified images for better understanding.

3.     In Fig. 5 caption, correct the error “2-ΔΔCT method”.

4.     Fig. 7, increase the font size of all figures and throughout the manuscript.

5.     Please provide the fluorescence images of stained cytoplasm and nucleus of the cells to understand the effect of EMTT to the cell’s morphology.

6.     Fig. 7(a and b), provide the scale bar.

7.     References are not consistent. Please make it consistent.

Comments on the Quality of English Language

Good 

Author Response

Comment 1: Fig. 3(e and f), increase the font size for better visibility.

Answer: Done.

Comment 2: Fig. (a and b), provide magnified images for better understanding.

Answer: Done.

Comment 3: In Fig. 5 caption, correct the error “2-ΔΔCT method”.

Answer: Done.

Comment 4: Fig. 7, increase the font size of all figures and throughout the manuscript.

Answer: Done. The front size has been increased for all figures. 

Comment 5: Please provide the fluorescence images of stained cytoplasm and nucleus of the cells to understand the effect of EMTT to the cell’s morphology.

Answer: Thank you for your suggestion. Unfortunately, we did not perform fluorescence staining of the cells in this study, so we are unable to provide the requested images. However, we are currently focusing on cell morphology changes after EMTT stimulation in ongoing live imaging projects, but at this stage, we are still at the initial phase.

Comment 6: Fig. 7(a and b), provide the scale bar.

Answer: Done.

Comment 7: References are not consistent. Please make it consistent.

Answer: The references have been updated in accordance with the author's instructions, using ACS style.